# SparseFormer: Detecting Objects in HRW Shots via Sparse Vision Transformer

## ABSTRACT

Recent years have seen an increase in the use of gigapixel-level image and video capture systems and benchmarks with high-resolution wide (HRW) shots. However, unlike close-up shots in the MS COCO dataset, the higher resolution and wider field of view raise unique challenges, such as extreme sparsity and huge scale changes, causing existing close-up detectors inaccuracy and inefficiency. In this paper, we present a novel model-agnostic sparse vision transformer, dubbed SparseFormer, to bridge the gap of object detection between close-up and HRW shots. The proposed SparseFormer selectively uses attentive tokens to scrutinize the sparsely distributed windows that may contain objects. In this way, it can jointly explore global and local attention by fusing coarse- and fine-grained features to handle huge scale changes. SparseFormer also benefits from a novel Cross-slice non-maximum suppression (C-NMS) algorithm to precisely localize objects from noisy windows and a simple yet effective multi-scale strategy to improve accuracy. Extensive experiments on two HRW benchmarks, PANDA and DOTA-v1.0, demonstrate that the proposed SparseFormer significantly improves detection accuracy (up to 5.8%) and speed (up to 3×) over the state-of-the-art approaches.

## CCS CONCEPTS

• Computing methodologies → Computer vision; • Computer systems organization → Embedded systems.

## KEYWORDS

Object Detection, Gigapixel, Super High-Resolution, Sparse Representation

ACM Reference Format:
Anonymous Author(s). 2018. SparseFormer: Detecting Objects in HRW Shots via Sparse Vision Transformer. In *Proceedings of Make sure to enter the correct conference title from your rights confirmation emai (Conference acronym 'XX).* ACM, New York, NY, USA, 9 pages. https://doi.org/XXXXXXX.XXXXXXX

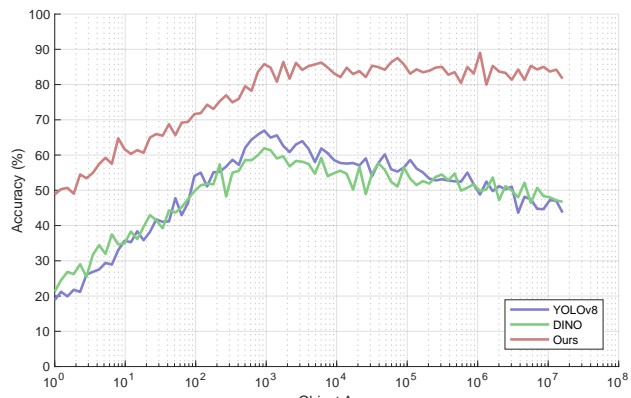

**Figure 1: Performance comparison in terms of object size on the PANDA dataset [43]. The horizontal axis indicates object sizes (the area of bounding boxes) on a logarithmic scale. The vertical axis shows detection accuracy per size. Both YOLOv8 [20] and DINO [54] underperform in handling extreme scale variations, especially for small and large objects. The proposed method performs well, achieving new state-of-the-art detection accuracy.**

## 1 INTRODUCTION

Object detection has been a challenging yet fundamental task in computer vision for the last decade. Close-up settings such as MS COCO [24] have shown impressive performance with successful real-world applications. However, with the development of imaging systems and new application requirements like UAVs, detecting objects in high-resolution wide (HRW) shots with square-kilometer scenes and gigapixel-level resolutions have drawn increasing attention [5, 10, 15, 22, 29, 30, 52, 53].

Detecting objects in HRW shots using close-up detectors is not effective due to several unique characteristics of HRW shots, as found in PANDA [43] and DOTA [45], compared to close-up shots like MS COCO. The most significant challenge is the sparse information in HRW shots, where objects often cover less than 5% of the image. This makes it difficult for detectors to extract key features from a sea of background noise, resulting in false positives within the background and false negatives within the object areas during training and testing. The second challenge is the varying scales of objects in HRW shots, with changes up to 100 times. Detectors relying on fixed settings of the receptive field and anchors cannot adapt to these extreme scales, as shown in Figure 1. For instance, YOLOv8 [20] underperforms in detecting small objects. While DINO [54] shows marginal improvement, it still falls short in adapting to such exaggerated scale changes, resulting in subpar detection of larger objects (Figure 2). Additionally, the typical two-stage downsampling schemes [5, 10, 21, 29] miss more small objects. The slicing strategy [1] can result in incomplete boxes when using NMS to merge prediction boxes, as shown in Figure 5. Therefore, it is imperative to bridge the gap between object detection in close-up and HRW shots.

Motivated by recent advanced techniques [28, 31, 37, 40, 41, 47] to enhance object detection accuracy, we present a novel detector

for HRW shots that employs a sparse Vision Transformer, called SparseFormer. SparseFormer uses attentive tokens selectively to concentrate on regions of an image where objects are sparsely distributed, facilitating the extraction of fine-grained features. To achieve this, it learns a ScoreNet to assess the importance of regions. By examining the variance of importance scores of all regions, our SparseFormer prioritizes regions that capture rich fine-grained details. In this way, it can focus on complex image regions rather than less significant ones (e.g., smooth content from the background). Concurrently, it divides each HRW shot into non-overlapping windows to extract coarse-grained features. Sharing a similar spirit with the receptive field strategy of the original Vision Transformer [8], our proposed SparseFormer combines coarse and fine-grained features, achieving much higher efficiency than Swin Transformer [26]. This greatly helps to handle large scale variations and detect both large and small objects accurately.

We further present two innovative techniques to improve detection accuracy against huge scale changes. First, we observe that conventional NMS refers to confidence scores only to merge detection results, leading to incomplete bounding boxes on oversized objects. To address this, we propose a novel Cross-slice NMS scheme (C-NMS) that favors large bounding boxes with high confidence scores. The proposed C-NMS scheme greatly improves the detection accuracy of oversized objects. Second, we use a multi-scale strategy to extract coarse-grained and fine-grained features. The multi-scale strategy enlarges the receptive field, enhancing the detection accuracy on both large and small objects. In summary, the main contributions of this work are as follows:

- We propose a novel sparse Vision Transformer based detector to handle huge scale changes in HRW images.
- We further use cross-window NMS and multi-scale schemes to improve detection on large and small objects.
- We extensively validate our method on two large-scale HRW-shot benchmarks, PANDA and DOTA-v1.0. Our method advances state-of-the-art performance by large margins.

## 2 RELATED WORK

**Close-up shot detection models.** The majority of common object detection datasets, such as PASCAL VOC [9] and MS COCO [24], collect high-resolution images with close-up shots, which has greatly contributed to the development of object detection. Based on the detection head, the literature can be broadly categorized into two types: one-stage detectors and two-stage detectors. The primary objective of the two-stage object detection is accuracy, and it frames the detection as a "coarse-to-fine" process [3, 12, 13, 18, 34]. On the other hand, one-stage detectors have an edge in terms of speed, such as YOLO [32]. Subsequent works have attempted to make improvements such as more anchors, better architecture, and richer training techniques [11, 25, 33]. To sum up, the current detectors exhibit great speed and accuracy in close-up shots.

**High-resolution wide shot detection models.** The introduction of imaging systems led to the development of a new benchmark for gigapixel-level detection with HRW shots called PANDA [43]. This benchmark has recently gained a lot of attention. Previous works on gigapixel-level detection focus on achieving lower latency through patch selection or arrangement [5, 10, 29]. However, they are unable

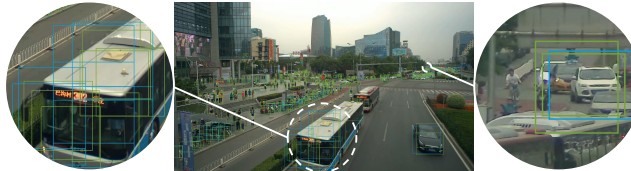

**Figure 2: Featured detection example on PANDA. The state-of-the-art detectors, YOLOv8 [20] (blue) and DINO [54] (green), relying on fixed settings of the receptive field and anchors yield incomplete bounding boxes on a large bus and miss detections on a small car.**

to solve the unique challenges faced in HRW shots. Some works use sparse policies on patches [31], self-attention heads [28], and transformer blocks [28] for image classification. PnP-DETR [41] exploits a poll and pool sampler to extract image features from the backbone and feed the sparse tokens to the attention encoder. This approach shows to be effective for object detection, panoptic segmentation, and image recognition. However, the sparse sampling on the backbone has not been adequately studied yet. DGE [37] is a plugin for vision transformers, but it is not flexible enough to be extended to ConvNet-based models or use arbitrary-size images as input. Therefore, how to design a flexible and model-agnostic architecture for object detection in HRW shots remains underexplored.

**Transformer backbones.** Transformers have been successful in natural language processing (NLP), and their potential for vision tasks has gained considerable attention. One such example is the Vision Transformer (ViT) [8], which uses a pure Transformer model for image classification and has shown promising results. However, ViT's computational costs for processing high-resolution images are impractical. Several methods have been attempted to reduce ViT model costs, including window-based attention [26], downsampling in self-attention [42, 44], and low-rank projection attention [46]. Other works use sparse policies on patches [31], self-attention heads [28], and transformer blocks [28] for image classification. Unfortunately, these methods suffer from significant accuracy drops when detecting objects in high-resolution wide shots.

## 3 PROPOSED METHOD

We address the unique challenges of HRW detection by proposing the Sparse Vision Transformer. This model efficiently extracts valuable features from sparse information, while enlarging the receptive field to handle huge scale changes. To tackle the problem of incomplete large objects on intersecting sliced areas, we modify vanilla NMS. Additionally, we introduce our HRW-based augmentation for both training and inference to enhance the detection accuracy for both large and small objects. The pipeline for inference using our proposed modules is shown in Figure 3.

### 3.1 Overview of SparseFormer

An ideal vision model should be able to extract meaningful information from sparse data using limited calculations, just like our human eyes tend to focus on valuable areas over unimportant background information. To achieve this, we design a novel Sparse

**Figure 3: Pipeline of SparseFormer in one forward inference. First, we perform multi-scale slicing on a gigapixel image. Then, we apply patch partitioning to each slice, and group neighboring patches into windows. Global Attention utilizes aggregated features to quickly obtain coarse-grained information. Local Attention selects important windows to extract fine-grained information.**

Vision Transformer called SparseFormer. It dynamically selects key regions and enables dynamic receptive fields to cover objects with various scales. The overall framework of SparseFormer is illustrated in Figure 4. Inspired by Swin Transformer [26], we split the input image into non-overlapping patches to generate tokens. Sparse-Former consists of four stages that work together to produce an adaptive representation. Each stage begins with a patch merging layer that concatenates the features of each group of $2 \times 2$ neighboring patches. The concatenated features are then projected to half of their dimension using a linear layer.

Each stage of SparseFormer is centered around attention blocks that are designed to capture long-range and short-range interactions at different scales. To achieve this, we take both the advantages of the vanilla self-attention Transformer block and the Swin Transformer block. In this way, we develop two distinct types of sparse-style blocks. One is used to capture long-range interactions at a coarse grain, while the other focuses on short-range interactions at a finer scale. To facilitate this approach, we introduce the concept of a *Window* which divides each feature map into equally spaced windows. Operations within each window are considered "local", while operations that encompass all windows are designated as "global."

We outline the global and local attention blocks in more detail. We construct the global block using the standard multi-head self-attention (MSA) [38] and MLP module with aggregated features, or only convolution layers, as detailed in Section 3.2. We construct the local block by adding a sparsification step and an inverse sparsification step before and after the Swin Transformer [26] block, as described in Section 3.3. Unlike previous work [41, 49], we do not build separate branches for global and local attention. Instead, the local attention is positioned after the global one to obtain more details, rather than different features. When a stage has multiple blocks, the ordering of global attention blocks (G) and local attention blocks (L) follows a pattern of 'GGLL'.

## 3.2 Global Attention on Aggregated Features

**Feature aggregation.** Global attention aims to capture coarse-grained features based on long-range interaction. As such, we generate low-resolution information by sparsifying the feature in each window. As shown in Figure 4, we begin each stage with the global attention block. The primary function of this block is to aggregate the features of each window. To achieve this, we take the input feature map $z$ and divide it into windows of size $M$, ensuring they do not overlap. The left-top location of each window is given by $(x, y)$, and each token within the window has a relative location $(\Delta x, \Delta y)$. We then calculate the aggregated features using the following formula:

$$\bar{z}_{x',y'} = \frac{\sum_{\Delta x, \Delta y} \alpha_{\Delta x, \Delta y} \cdot z_{x+\Delta x, y+\Delta y}}{\sum_{\Delta x, \Delta y} \alpha_{\Delta x, \Delta y}}, \qquad (1)$$

Here, $x' = x/M$ and $y' = y/M$. $\alpha_{\Delta x, \Delta y}$ is the weight of each token. In this paper, we assign equal weights to all tokens by setting $\alpha_{\Delta x, \Delta y} = 1$. After aggregating the features using the above formula, we obtain the aggregated feature $\bar{z}$ which can be further used for attention.

**Window-level global attention.** Feature aggregation is a technique that reduces the number of tokens by a multiple of $M^2$, which is equivalent to a $M\times$ downsampling of resolution. This reduction in tokens allows us to use global attention interaction without expensive computation. With the aggregated features, consecutive global blocks are computed as follows:

$$\bar{z}^l = \text{Layer}(\bar{z}^{l-1}; \Theta), \qquad (2)$$

where $\bar{z}^l$ refers to the output features of the $l$-th global block.

**Inverse aggregated features.** The aggregated features contain abstract information that facilitates global content-dependent interactions among different image regions. However, their resolution differs from the input feature map. Consequently, we convert the window-level features back to the token-level using an inverse function of equation Equation (1), as follows:

$$z_{x+\Delta x, y+\Delta y} = \alpha_{\Delta x, \Delta y} \cdot \bar{z}_{x',y'}. \qquad (3)$$

Here, $x = M \cdot x'$ and $y = M \cdot y'$, where $(x', y')$ and $(x, y)$ represent the location on the input and output feature maps, respectively. Additionally, $(\Delta x, \Delta y)$ represents the relative location with respect to $(x, y)$. We consider $(x, y)$ as the top-left of each window on the output feature map, where the windows are partitioned in the same manner as in the feature aggregation process.

This step extracts the output feature map from the successive global block (Equation (2)). Then, we invert it using Equation (3) and denote the resulting feature map by $z^{\text{global}}$. It is worth noting that the final global feature $z^{\text{global}}$ has the same resolution as the input feature map $z$. Even though aggregated features have a lower resolution, the global attention operation can provide more non-local information with little extra computation.

## 3.3 Local Attention on Sparse Windows

**Variance-based scoring.** Note that coarse-grained features for each window can achieve high efficiency. However, we still need fine-grained features that can extract object details to accurately

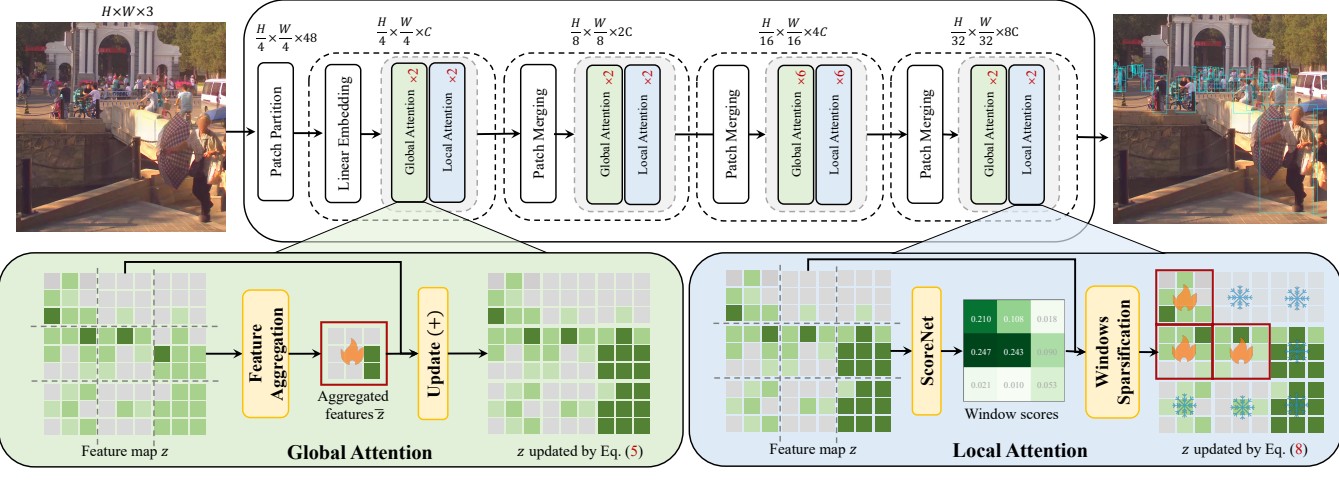

**Figure 4: Network Architecture of SparseFormer. The** **red box** **represents the interaction range of attention.🔥 means tokens are updated by self-attention and ❄ means they remain unchanged. We partition the image into tokens and group them into windows. Global attention extracts coarse-grained features from all windows based on aggregated tokens and merges them with the original features. Local attention selects only the windows with complex details for fine-grained feature extraction through our ScoreNet, while the rest retain their original features to save computational resources.**

detect objects. As such, we drop certain windows based on their low information content to reduce computation. Our goal is to identify windows that require further local attention because their window-level feature cannot represent their inner token-level features.

We begin with an initial feature map $z$, which is before global and local attention, and has dimensions of $H$, $W$, and $C$. We then get the aggregated feature $\bar{z}$ from $z$ using Equation (1) and apply the inverse sparsification function via Equation (3), which produces an intermediate feature map $\hat{z}$ with the same resolution as $z$. Next, we calculate the residual $r$ between $\hat{z}$ and $z$ and concatenate the features of each window to obtain the tokens of $M \times M \times C$ dimensions that are $\frac{H}{M} \times \frac{W}{M}$ in size. We can easily calculate the variance of each window from these tokens. We construct a ScoreNet using MLP to add learnable weights to each residual. The score of each window is given by:

$$\text{ScoreNet}(z, \hat{z}) = \text{SoftMax}(\text{MLP}(z - \hat{z})), \quad (4)$$

where the MLP projects $(M \times M \times C)$-dimensional features for each window to 1 dimension, and the SoftMax operation calculates the score for each window. A higher score indicates greater variance, meaning high-variance windows require fine-grained attention. In other words, we discard windows with lower scores during local attention. Once we have ranked the windows, we can selectively choose a part of them to generate finer-level features. Before doing so, we update the feature map $z$ with global feature $z^{\text{global}}$ using:

$$z \leftarrow z + z^{\text{global}}. \quad (5)$$

**Windows sparsification.** We start by analyzing the global attention and variance-based scoring to obtain the initial feature $z$ and scores for each window. Next, we partition $z$ into windows of size $\frac{H}{M} \times \frac{W}{M}$, in the same way as the ScoreNet. We represent these windows as a matrix $Z \in \mathbb{R}^{N \times D}$, where $N$ is the total number of windows, i.e., $N = \frac{H}{M} \times \frac{W}{M}$ and $D = M \times M \times C$. To determine

which windows to keep, we define a hyperparameter $k$ to represent the keeping ratio. We maintain a binary decision mask vector $A \in \{0, 1\}^N$ to indicate whether to drop or keep each window based on $k$ and scores. The value of $k$ would depend on the specific task at hand, and can be adjusted as required. The sparse matrix $S \in \mathbb{R}^{K \times N}$ collects the one-hot encoding of vector $A$, where $K$ is the number of keeping windows, i.e., $K = k \cdot N$. Using this sparse matrix, we compute the features of the sparse windows as follows:

$$\hat{Z}_s = S \times Z, \quad (6)$$

The output feature $\hat{Z}_s \in \mathbb{R}^{K \times D}$ is then used as input for the local attention.

**Shifted window-based attention.** We utilize the Shift Window-based Attention module which was first introduced in Swin Transformer [26]. The consecutive local blocks can be represented as:

$$\begin{aligned} z^l &= \text{W-Layer}(z^{l-1}; \Theta), \\ z^{l+1} &= \text{SW-Layer}(z^l; \Theta), \end{aligned} \quad (7)$$

where $z^l$ denotes the output features of the local block $l$. The layer can be either a self-attention or convolution module. To fuse the output and input features of local attention, we use the following equation:

$$\hat{Z} \leftarrow A^T \times \hat{Z}_s + (\mathbf{1} - A^T) \times Z. \quad (8)$$

Here, $\hat{Z}_s$ is updated by local attention, while $\hat{Z}$ is the output of each stage of SparseFormer. Finally, we revert $\hat{Z}$ back into the original dimensional space of $H \times W \times C$ to obtain the final feature map, denoted as $z$. The window-based attention, based on variance-based scoring, can extract more local information in a lightweight form, thus improving the detection performance of small objects while saving computation for the background.

---

**Algorithm 1:** Cross-slice NMS (C-NMS)

**Variables :** $\mathcal{B}_1, \mathcal{B}_2$ are candidate box sets from two slices, $\tau$
  is the C-NMS threshold;
**Functions:** NMS($\cdot$) is the conventional NMS;
  AREA($\cdot$) calculates the area of a box;

1 $\mathcal{B}_1' \leftarrow$ NMS($\mathcal{B}_1$); $\mathcal{B}_2' \leftarrow$ NMS($\mathcal{B}_2$);
2 $\mathcal{B} \leftarrow \mathcal{B}_1' \cup \mathcal{B}_2'$; $\mathcal{B}' \leftarrow \emptyset$;
3 **while** $\mathcal{B} \neq \emptyset$ **do**
4    $m \leftarrow \arg\max_i$ AREA($b_i$), s.t. $b_i \in \mathcal{B}$;
5    $\mathcal{B}' \leftarrow \mathcal{B}' \cup \{b_m\}$; $\mathcal{B} \leftarrow \mathcal{B} - \{b_m\}$;
6    **for** $b_i \in \mathcal{B}$ **do**
7      **if** $IoU(b_i, b_m) \geq \tau$ **then**
8        $\mathcal{B} \leftarrow \mathcal{B} - \{b_i\}$;

9 **return** $\mathcal{B}'$;

---

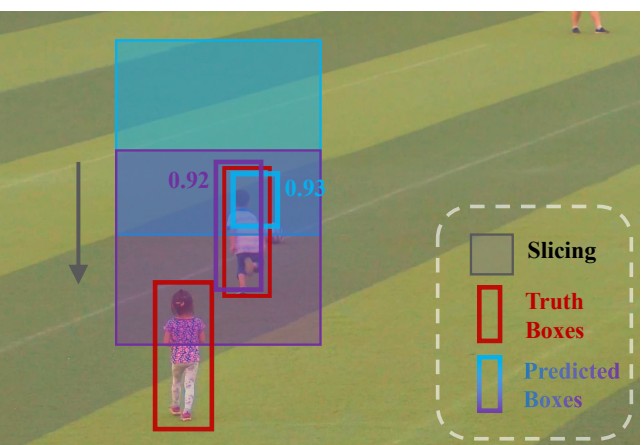

**Figure 5: Featured detection example on large objects with slicing aid. Detector yields two boxes based on overlapped slices. NMS, relying on the detection scores, will wrongly select the blue box for the kid.**

**End-to-end Optimization.** It is challenging to optimize the ScoreNet because we only use the output to sort the windows, and the gradient cannot be back-propagated. To overcome this issue, we implement the Gumbel-Softmax trick [27] to relax the sampling process, making it differentiable. This trick provides a bridge for gradient back-propagation between soft values and binarized values through re-parameterization. Hence, we re-write Equation (5) as:

$$z \leftarrow z + (1-s) \times z^{\text{global}}, \qquad (9)$$

Here, $s$ represents the output of the SoftMax function, which indicates the scores of windows.

### 3.4 Cross-slice Non-Maximum Suppression

In HRW shot processing, the slicing strategy generates box candidates for each slice, which must then be merged into a mutually non-conflicting box set. However, using Non-Maximum Suppression (NMS) to select the highest-scoring boxes may lead to incomplete boxes when objects are on the edge regions of multiple slices

(For a more detailed explanation and visual representation, refer to Figure 5). To address this, we propose a Cross-slice Non-Maximum Suppression (C-NMS) strategy, as shown in Algorithm 1, that prioritizes boxes with the maximum area across multiple slices, rather than just the highest scores. The C-NMS algorithm consists of two stages: a local suppression stage and a cross-slice suppression stage.

### 3.5 Multi-scale Training and Inference

Due to memory limitations, it is not possible to train and test super high-resolution datasets at their original size. Therefore, we use a slicing strategy in both the training and testing phases. To make better use of the multi-scale information, we use high-resolution images and divide them into slices of varying sizes using the slicing strategy. All slices are scaled to the same size, enabling effective training and inference for the object detector. We divide the image into grids of 16×16, 8×8, 4×4, and 2×2 grids, respectively, and remove the slices with no objects. This approach allows us to analyze and understand the complex features of these images, ultimately improving the overall accuracy and effectiveness of the detector.

During the inference phase, we use slicing windows of two sizes: the original one and one quarter of both height and width. Instead of simply combining the two windows, we set different receptive fields for the two types of windows with a threshold $T_a$. Based on the first window, we remove the prediction boxes larger than $T_a$. We only keep the boxes larger than $T_a$ for the second window. This follows the idea of scale-specific design [35, 36], where we should arrange each window to cover the appropriate scale to improve performance. With this technique, we can quickly and accurately process high-resolution images.

## 4 EXPERIMENT

### 4.1 Effectiveness Evaluation

**Datasets.** Our evaluation is on two public benchmarks with HRW shots, PANDA [43] and DOTA-v1.0 [45]. PANDA is the first human-centric gigapixel-level dataset. It contains 18 scenes with over 15,974.6k bounding boxes annotated. Specifically, there are 13 scenes for training and 5 scenes for testing. DOTA is a large-scale dataset to evaluate the oriented object detection in aerial images with sizes up to 4000×4000. It contains 2,806 images and 188,282 instances with oriented bounding box annotations, covered by 15 object classes.

**Evaluation metrics.** We report the FLOPs and standard COCO metrics including $AP_{total}$, $AP_S$ (< 96×96), $AP_M$ (96×96−288×288) and $AP_L$ (> 288×288). For quantitative efficiency evaluation, we use the average FLOPs of each detector to process a 1280 × 800 window in the datasets. Further, we calculate FLOPs in foreground and background, respectively, to show the efficiency of our SparseFormer to reduce the computation on backgrounds.

**Implementation details.** We implement the detectors using MMDetection [4], incorporating both the officially provided DINO and Dynamic-Head, and integrating the open-source code for the DEG backbone. To ensure a fair comparison, we evaluate these two detectors across four different backbones, including Swin, DEG, and our own proprietary design, all configured with identical numbers of hyperparameters (e.g., depths, embedding dimension, number

**Table 1: Comparison with the state-of-the-arts on PANDA. "F" and "B" denote foreground and background, respectively (A=F+B). "*" denotes the re-implementation in [10]. The GFLOPs on the two-stage detector do not include the detection heads due to the dynamic cost.**

| Method | Backbone | GFLOPs-F | GFLOPs-B | GFLOPs-A | $AP_{Total}$ | $AP_S$ | $AP_M$ | $AP_L$ |
|---|---|---|---|---|---|---|---|---|
| FasterRCNN [34] | ResNet-101 | 14.15 | 268.98 | 283.14 | - | 0.190 | 0.552 | 0.744 |
| FasterRCNN* [10] | ResNet-50 | 10.35 | 196.71 | 207.07 | 0.705 | 0.203 | 0.712 | 0.760 |
| RetinaNet [23] | ResNet-101 | 15.77 | 299.62 | 315.39 | - | 0.221 | 0.561 | 0.740 |
| CascadeRCNN [3] | ResNet-101 | 15.54 | 295.24 | 310.78 | - | 0.227 | 0.579 | 0.765 |
| ClusDet [48] | ResNet-50 | 10.35 | 196.71 | 207.07 | 0.718 | 0.219 | 0.696 | 0.782 |
| DMNet [21] | ResNet-50 | 10.35 | 196.71 | 207.07 | 0.540 | 0.119 | 0.371 | 0.714 |
| GigaDet [5] | CSP-DarkNet-53 | 4.61 | 87.59 | 92.20 | 0.684 | 0.210 | 0.599 | 0.762 |
| PAN [10] | ResNet-50 | 10.35 | 196.71 | 207.07 | 0.715 | 0.256 | 0.719 | 0.768 |
| Dynamic-Head [6] | Swin-T | 5.74 | 109.1 | 114.8 | 0.592 | 0.165 | 0.537 | 0.694 |
| Dynamic-Head+DEG [37] | PVT-DEG | 6.12 | 60.11 | 66.23 | 0.575 | 0.154 | 0.508 | 0.695 |
| Dynamic-Head+Ours | SparseFormer | 6.29 | 58.35 | 64.64 | 0.771 | 0.364 | 0.740 | 0.863 |
| DINO [54] | Swin-T | 6.64 | 126.19 | 132.84 | 0.606 | 0.367 | 0.612 | 0.649 |
| DINO+DEG [37] | PVT-DEG | 6.77 | 78.57 | 85.34 | 0.582 | 0.339 | 0.578 | 0.624 |
| DINO+Ours | SparseFormer | 6.90 | 68.81 | 75.71 | 0.780 | 0.508 | 0.781 | 0.823 |
| DINO [54] | ResNet-50 | 6.21 | 118.02 | 124.24 | 0.542 | 0.289 | 0.530 | 0.592 |
| DINO+Ours | SparseNet | 6.53 | 100.97 | 107.50 | 0.746 | 0.381 | 0.754 | 0.797 |

**Table 2: Ablation studies on component effectiveness.**

| | $AP_{Tot.}$ | $AP_S$ | $AP_M$ | $AP_L$ |
|---|---|---|---|---|
| Swin Block | 0.577 | 0.309 | 0.594 | 0.620 |
| +MS Train | 0.590 | 0.344 | 0.599 | 0.628 |
| +MS Inference | 0.606 | 0.367 | 0.612 | 0.649 |
| Local Block | 0.594 | 0.319 | 0.583 | 0.644 |
| +Global Block | 0.654 | 0.238 | 0.611 | 0.744 |
| +MS Train | 0.765 | 0.386 | 0.763 | 0.817 |
| +MS Inference | 0.773 | 0.443 | 0.775 | 0.813 |
| +C-NMS | 0.780 | 0.508 | 0.781 | 0.823 |

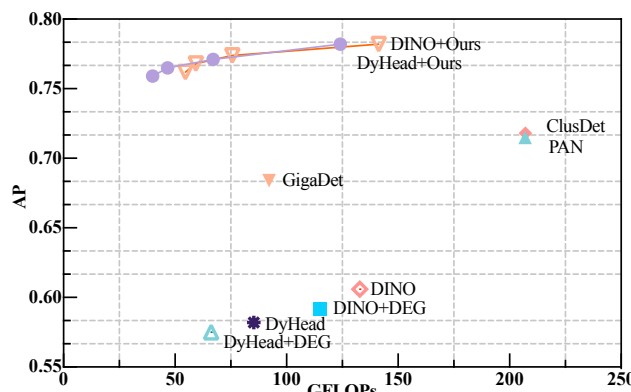

**Figure 6: FLOPs vs. AP. Our methods reduce FLOPs up to 50% and improve detection accuracy in HRW shots.**

of multi-heads). All models are trained from scratch for 36 epochs, in line with the observations in [17]. This also ensures a fair comparison with Swin and DEG, affirming that the performance improvement stems from our novel design rather than from better pre-trained weights.

**Results on PANDA.** We compare our model with a different keeping ratio $k$ to current state-of-the-art methods on the first gigapixel-level dataset PANDA, which not only has the challenge of wide FoV, but also super high resolution. The results are presented in Table 1. We first produce two baselines, one based on ATSS framework [55] with dynamic head block [6] which achieves GFLOPs of 114.80, the other based on DINO [54] and GFLOPs of 132.84. Then, the backbone is modified to SparseFormer for further experiments. Note that the keeping ratio means the ratio of keeping tokens based on the previous stage, so the ratio of each stage based on the full number of tokens is $[k, k^2, k^3, k^4]$. We can observe our method achieves more than 5% increase in AP over SotAs, with only 75.71 GFLOPs (43% reduction from Swin-T, 63% reduction from PAN [10]). The most notable thing is that the reduced FLOPs are mainly from the background region, which is why we can significantly reduce the amount of computation but maintain high performance. An additional note is that GigaDet and PAN is to accelerate the detector by optimizing the process. Unlike these approaches, our work does not prescribe a specific pipeline. Instead, we propose a model-agnostic strategy that could be seamlessly integrated into existing pipelines.

**Results on DOTA.** We choose aerial images that also have HRW shots to verify the generalization. The compared methods include: Faster RCNN-O [34], ICN [2], RoI-Transformer [7], CADNet [53], DRN [30], CenterMap [39], SCRDet [52], $R^3$Det [51], $S^2$A-Net [15], CFA [14], CSL [50], ReDet [16], Or-RepPoints [22]. RoI-Transformer is used as the baseline detector for comparison and we set $k = 0.7$. In Table 3, SparseFormer improves mAP from 69.56% to 77.94% and reduces 296.74 GFLOPs to 174.31 GFLOPs. Compared to SotA

**Table 3: Comparison with the state-of-the-art methods on DOTA-v1.0. The short names for categories are defined as (abbreviation-full name): PL-Plane, BD-Baseball diamond, BR-Bridge, GTF-Ground field track, SV-Small vehicle, LV-Large vehicle, SH-Ship, TC-Tennis court, BC-Basketball court, ST-Storage tank, SBF-Soccer-ball field, RA-Roundabout, HA-Harbor, SP-Swimming pool, and HC-Helicopter. BB means Backbone. Conv. means our SparseNet. Trans. means our SparseFormer.**

| Method | BB | PL | BD | BRe | GTF | SV | LV | SH | TC | BC | ST | SBF | RA | HA | SP | HC | mAP | GFLOPs |
|---|---|---|---|---|---|---|---|---|---|---|---|---|---|---|---|---|---|---|
| FR-O | R-101 | 79.42 | 77.13 | 17.70 | 64.05 | 35.30 | 38.02 | 37.16 | 89.41 | 69.64 | 59.28 | 50.30 | 52.91 | 47.89 | 47.40 | 46.30 | 54.13 | 282.88 |
| ICN | R-101 | 81.36 | 74.30 | 47.70 | 70.32 | 64.89 | 67.82 | 69.98 | 90.76 | 79.06 | 78.20 | 53.64 | 62.90 | 67.02 | 64.17 | 50.23 | 68.16 | - |
| RoI-Trans. | R-101 | 88.64 | 78.52 | 43.44 | 75.92 | 68.81 | 73.68 | 83.59 | 90.74 | 77.27 | 81.46 | 58.39 | 53.54 | 62.83 | 58.93 | 47.67 | 69.56 | 296.74 |
| CADNet | R-101 | 87.80 | 82.40 | 49.40 | 73.50 | 71.10 | 63.50 | 76.60 | 90.90 | 79.20 | 73.30 | 48.40 | 60.90 | 62.00 | 67.00 | 62.20 | 69.90 | - |
| DRN | H-104 | 88.91 | 80.22 | 43.52 | 63.35 | 73.48 | 70.69 | 84.94 | 90.14 | 83.85 | 84.11 | 50.12 | 58.41 | 67.62 | 68.60 | 52.50 | 70.70 | - |
| CenterMap | R-50 | 88.88 | 81.24 | 53.15 | 60.65 | 78.62 | 66.55 | 78.10 | 88.83 | 77.80 | 83.61 | 49.36 | 66.19 | 72.10 | 72.36 | 58.70 | 71.74 | - |
| SCRDet | R-101 | 89.98 | 80.65 | 52.09 | 68.36 | 68.36 | 60.32 | 72.41 | 90.85 | 87.94 | 86.86 | 65.02 | 66.68 | 66.25 | 68.24 | 65.21 | 72.61 | - |
| R$^3$Det | R-152 | 89.49 | 81.17 | 50.53 | 66.10 | 70.92 | 78.66 | 78.21 | 90.81 | 85.26 | 84.23 | 61.81 | 63.77 | 68.16 | 69.83 | 67.17 | 73.74 | 480.33 |
| S$^2$A-Net | R-50 | 89.11 | 82.84 | 48.37 | 71.11 | 78.11 | 78.39 | 87.25 | 90.83 | 84.90 | 85.64 | 60.36 | 62.60 | 65.26 | 69.13 | 57.94 | 74.12 | 193.11 |
| CFA | R-101 | 89.26 | 81.72 | 51.81 | 67.17 | 79.99 | 78.25 | 84.46 | 90.77 | 83.40 | 85.54 | 54.86 | 67.75 | 73.04 | 70.24 | 64.96 | 75.05 | 265.96 |
| CSL | R-152 | 90.25 | 85.53 | 54.64 | 75.31 | 70.44 | 73.51 | 77.62 | 90.84 | 86.15 | 86.69 | 69.60 | 68.04 | 73.83 | 71.10 | 68.93 | 76.17 | 383.13 |
| ReDet | ReR-50 | 88.79 | 82.64 | 53.97 | 74.00 | 78.13 | 84.06 | 88.04 | 90.89 | 87.78 | 85.75 | 61.76 | 60.39 | 75.96 | 68.07 | 63.59 | 76.25 | - |
| O-Rep. | Swin-T | 89.11 | 82.32 | 56.71 | 74.95 | 80.70 | 83.73 | 87.67 | 90.81 | 87.11 | 85.85 | 63.60 | 68.60 | 75.95 | 73.54 | 63.76 | 77.63 | 221.32 |
| Ours | Conv. | 89.00 | 82.42 | 55.04 | 74.19 | 79.62 | 81.54 | 87.77 | 90.90 | 87.08 | 85.83 | 64.18 | 64.13 | 74.65 | 71.21 | 58.74 | 76.42 | **167.24** |
| Ours | Trans. | 89.45 | 85.81 | 55.18 | 77.65 | 78.51 | 83.45 | 87.81 | 90.90 | 86.88 | 86.26 | 63.59 | 67.30 | 75.94 | 73.65 | 66.69 | **77.94** | 174.31 |

**Table 4: Ablation studies on sparse ratio. Based on the following results, we set the ratio to 0.7 to maintain the best balance between speed and accuracy in other experiments.**

| Method | Ratio | GFLOPs | | | AP | | | |
|---|---|---|---|---|---|---|---|---|
| | | Fore | Back | All | Total | Small | Medium | Large |
| DyHead | 0.1 | 5.91 | 31.24 | 37.15 | 75.1 | 31.6 | 71.4 | 85.4 |
| | 0.3 | 5.90 | 34.14 | 40.04 | 75.9 | 34.5 | 72.7 | 85.3 |
| | 0.5 | 6.16 | 40.59 | 46.75 | 76.5 | 34.7 | 73.5 | 85.5 |
| | 0.7 | 6.19 | 60.87 | 67.06 | 77.1 | 36.4 | 74.0 | 86.3 |
| | 1.0 | 6.21 | 117.9 | 124.1 | 78.2 | 44.2 | 75.4 | 86.3 |
| DINO | 0.1 | 6.74 | 44.77 | 51.51 | 75.6 | 33.5 | 75.1 | 82.4 |
| | 0.3 | 6.79 | 48.86 | 54.65 | 76.1 | 41.1 | 76.2 | 81.3 |
| | 0.5 | 6.84 | 52.53 | 59.37 | 76.7 | 42.5 | 77.2 | 80.9 |
| | 0.7 | 6.90 | 68.81 | 75.71 | 77.3 | 44.3 | 77.5 | 81.3 |
| | 1.0 | 7.06 | 134.2 | 141.3 | 78.0 | 53.0 | 77.8 | 81.9 |

Transformer-based method Or-RepPoints, we achieve 0.3% AP improvement and reduce 21% GFLOPs. Compared to the method with similar computation S$^2$A-Net, we surpass its AP by 3.82%. This indicates a significant enhancement in terms of accuracy and efficiency. The DOTA [45] dataset presents a formidable challenge, yet our approach achieves the precision of the current SotAs with significantly fewer FLOPs. This not only validates the design intention behind SparseFormer to reduce computational demands but also demonstrates its generalizability across various tasks and domains.

## 4.2 Ablation Study

**Component effectiveness.** We investigate the effectiveness of global block, C-NMS, multi-scale training (MS Train) and inference (MS Inference). Evaluation is conducted on the PANDA with $k = 0.7$. As shown in Table 2, all components can significantly improve performance with a little extra cost which also shows that our strategies are useful for object detection in HRW shots.

**Table 5: Ablation studies on the selection strategy. The results indicate that using the difference is more effective than directly inputting $z$, and using the p-norm brings no gains. Therefore, we choose $z - \hat{z}$ in other experiments.**

| | AP$_{Total}$ | AP$_S$ | AP$_M$ | AP$_L$ |
|---|---|---|---|---|
| $\hat{z}$ | 0.748 | 0.452 | 0.757 | 0.786 |
| $z$ | 0.735 | 0.283 | 0.737 | 0.805 |
| $z - \hat{z}$ | 0.773 | 0.443 | 0.775 | 0.813 |
| $|z - \hat{z}|$ | 0.773 | 0.474 | 0.770 | 0.818 |
| $(z - \hat{z})^2$ | 0.771 | 0.468 | 0.784 | 0.811 |

**Keeping ratio.** Our strategy involves discarding the grids that are deemed unimportant. We study the impact of the grid-keeping ratio on the final performance. Table 4 presents our findings, where the keeping ratio $k$ is represented as $[k, k^2, k^3, k^4]$ for each stage. As the features become sparser, we observe a significant reduction in FLOPs, but the decrease in accuracy is insignificant.

**Effect on ScoreNet.** We study the effect of different post-processing of residual values which feed into ScoreNet (introduced in Section 3.3). $z$ and $\hat{z}$ denote the original features and aggregated features, respectively, which are the same in Equation (4). As can be seen from the last three lines, several variants based on residuals have a performance within the error range, so we consider using less computation and do not perform any redundant processing on them. Compared with $Z$, which directly uses all the features in the window, the average feature $\hat{z}$ can achieve better results. We think this is because the ScoreNet is a simple MLP, it cannot take advantage of complex features $z$ well, while the $\hat{z}$ is easier to be classified based on color (blue for the sky and green for glasses) and other patterns.

## 4.3 Comparison on Edge Device

The HRW shots are usually captured by edge devices like UAVs. UAV detectors typically cannot run on large computing devices, but instead run on low-power edge devices. Because it is usually

**Figure 7: Visualization of the window scores. We illustrate the first-stage window scores of SparseFormer, with the left two columns of images from PANDA [43] and the right two columns from DOTA [45]. Highlighted points indicate areas requiring extraction of fine-grained features.**

**Table 6: Comparison of the inference time on edge-device.**

| Method | $AP_{Total}$ | $AP_S$ | $AP_M$ | $AP_L$ | Latency(s) | FPS |
|---|---|---|---|---|---|---|
| FasterRCNN [34] | - | 0.190 | 0.552 | 0.774 | 140 | 0.007 |
| CascadeRCNN [3] | - | 0.227 | 0.579 | 0.765 | 200 | 0.005 |
| PAN [10] | 0.715 | 0.256 | 0.719 | 0.768 | 43 | 0.023 |
| DyHead [6] | 0.592 | 0.165 | 0.537 | 0.694 | 63 | 0.015 |
| DyHead+DEG [37] | 0.575 | 0.154 | 0.508 | 0.695 | 58 | 0.017 |
| DyHead+Ours | 0.771 | 0.364 | 0.740 | 0.863 | 52 | 0.019 |
| DINO [54] | 0.606 | 0.367 | 0.612 | 0.649 | 19 | 0.052 |
| DINO+DEG [37] | 0.582 | 0.339 | 0.578 | 0.624 | 15 | 0.066 |
| DINO+Ours | 0.773 | 0.443 | 0.775 | 0.813 | 14 | 0.071 |

difficult to quantify FLOPs on edge devices, we use NVIDIA AGX Orin (top power 60W) to evaluate the average inference time of each detector on the gigapixel-level images from PANDA and the results are shown in Table 6. Notably, our method can largely reduce inference time compared to previous methods. Our method is 3× faster than PAN and has 5.8% increase of AP. We can see because of the complex head structure, the inference speed of dynamic-head [6] is not ideal. On the opposite, DINO [54] shows promising FPS than the previous work and the improvement of the speed is clearer. Compared to the competitive approach DEG [37], our method could achieve much better performance with faster speed.

### 4.4 Model-agnostic Study

It is noteworthy that our strategy is model-agnostic, enabling seamless integration with either ConvNet or Transformer architectures. This flexibility leads to the creation of SparseNet and SparseFormer. Building upon the previously mentioned SparseFormer, we have innovated by substituting every self-attention module with convolution layers. As illustrated in Table 1 and Table 3, SparseNet demonstrates performance that is not only comparable but also competitive with the renowned ResNet [19]. Especially noteworthy

is that SparseNet reduces the GFLOPs up to 56% while increasing accuracy compared to CSL [50] and it achieves the lowest GFLOPs on the DOTA dataset [45], underscoring its efficiency and effectiveness in complex computational tasks.

### 4.5 Visualization of Sparse Windows

In order to better understand how window sparsification works, we visualize the selected windows from each stage in Figure 7. The red patches represent regions with higher scores, while the blue patches indicate lower scores. SparseFormer will perform fine-grained feature extraction on the regions with higher scores. This illustration highlights the advantages of computation reduction on background areas and low-entropy foregrounds. Additionally, the results validate the effectiveness of our SparseFormer approach. The PANDA [43] and DOTA [45] datasets focus on different target objects, they share the common characteristic of containing large-scale background areas, making the sparsification approach particularly relevant. We believe that this methodology will not only benefit object detection in HRW shots but also various other vision tasks.

### 5 CONCLUSION

We introduced SparseFormer, a sparse Vision Transformer-based detector designed for HRW shots. It uses selective token utilization to extract fine-grained features and aggregate features across windows to extract coarse-grained features. The combination of fine and coarse granularity effectively leverages the sparsity of HRW shots, facilitating handling extreme scale variations. Our Cross-slice NMS scheme and multi-scale strategy help detect oversized and diminutive objects. Experiments on PANDA and DOTA-v1.0 benchmarks show significant improvement over existing methods, advancing state-of-the-art performance in HRW shot object detection.

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
