# OpenReview forum: "SparseFormer: Detecting Objects in HRW Shots via Sparse Vision Transformer"
_acmmm.org/ACMMM/2024/Conference — MM2024 Poster_

### Official Review · Reviewer_qjet · 2024-04-28

**Rating:** 4
**Confidence:** 1

**Summary:**

The paper introduces SparseFormer, a novel model-agnostic sparse vision transformer designed to address the challenges of object detection in high-resolution wide (HRW) shots. Unlike traditional close-up object detection methods, SparseFormer selectively uses attentive tokens to focus on sparsely distributed regions that may contain objects, enabling it to handle extreme scale variations and sparsity. The model employs a Cross-slice non-maximum suppression (C-NMS) algorithm for precise object localization and a multi-scale strategy to enhance accuracy. Experiments on two HRW benchmarks, PANDA and DOTA-v1.0, demonstrate that SparseFormer significantly improves detection accuracy and speed over state-of-the-art approaches.

**Strengths:**

+ The paper is well-structured.

+ The paper offers extensive experimental validation on two large-scale HRW-shot benchmarks, PANDA and DOTA-v1.0.

+ The inclusion of visualizations, such as the window scores from SparseFormer, helps in understanding how the model prioritizes regions for fine-grained feature extraction.

**Limitations:**

- The latex format is not up to date.

- No loss function is given for the overall network.

- In the experiments in Table 2, the APs for Local Block is 0.319, why does the performance drop with the addition of Global Block (from 0.319 to  0.238)?

- The paper needs to be checked for some errors, such as line 512, where equation 9 should end with a period.

**Suitability:**

2

---

### Official Review · Reviewer_xF4H · 2024-05-23

**Rating:** 4
**Confidence:** 3

**Summary:**

The paper addresses the challenges of object detection in high-resolution wide-view (HRW) images, such as extreme sparsity and large scale variations. Authors proposes SparseFormer, a sparse vision Transformer that uses selective attention tokens to focus on potential object regions in HRW images. Introduces a ScoreNet for importance evaluation, Cross-window Non-Maximum Suppression (C-NMS) for accurate large object detection, and a multi-scale strategy for enhanced detection. Extensive experiments on PANDA and DOTA-v1.0 datasets were conducted.

**Strengths:**

1. The proposed method is relatively novel with sound and meaningful motivation. The proposed idea of global-local sparse attention is particularly well-suited for addressing the problem.
2. The paper is well-organized and easy to follow.
3. Extensive experiments were conducted.

**Limitations:**

1. The comparison in the paper only includes methods up to 2022. Please include the latest work from the past two years.
2. ScoreNet is an existing technique, as proposed in DynamicViT (NeurIPS 2021).

**Suitability:**

2

---

### Official Review · Reviewer_W73g · 2024-05-24

**Rating:** 6
**Confidence:** 3

**Summary:**

The paper introduces SparseFormer, a model-agnostic sparse vision transformer for high-resolution wide (HRW) shots. SparseFormer uses attentive tokens to focus on sparsely distributed windows and combines global and local attention by fusing coarse- and fine-grained features. It also features a Cross-slice non-maximum suppression (C-NMS) algorithm and a multi-scale strategy to enhance accuracy. Experiments on HRW benchmarks PANDA and DOTA-v1.0 show significant improvements in detection accuracy (up to 5.8%) and speed (up to 3×) over current methods.

**Strengths:**

1. This paper is well-written and easy to follow. The motivation is clear and the explanation to sparseformer is concise and easy to understand.
2. Extensive experiments denonstrate the effectiveness  and efficiency of sparseformer, which is exciting and promising.
3. Further analyses towards sparseformer, including ablation studies and visualizations, seems convincing and inspiring.
4. From my perspective, sparseformer will be agreat contribution to HRW research community.

**Limitations:**

None in particular

**Suitability:**

3

---

### Meta-Review · Area_Chair_QMpm · 2024-07-01

**Recommendation:** Accept (Poster)
**Confidence:** 5

**Metareview:**

This paper presents a novel model-agnostic SparseFormer approach to bridge the gap of object detection between close-up and HRW shot. This authors addressed the concerns of most reviewers well and received all positive ratings after the rebuttal; thus, I recommend acceptance for this submission.